# Prevalence of Psychological Frailty in Japan: NCGG-SGS as a Japanese National Cohort Study

**DOI:** 10.3390/jcm8101554

**Published:** 2019-09-27

**Authors:** Hiroyuki Shimada, Sangyoon Lee, Takehiko Doi, Seongryu Bae, Kota Tsutsumimoto, Hidenori Arai

**Affiliations:** 1Center for Gerontology and Social Science, National Center for Geriatrics and Gerontology, Aichi 474-8511, Japan; sylee@ncgg.go.jp (S.L.); take-d@ncgg.go.jp (T.D.); bae-sr@ncgg.go.jp (S.B.); k-tsutsu@ncgg.go.jp (K.T.); 2National Center for Geriatrics and Gerontology, Aichi 474-8511, Japan; harai@ncgg.go.jp

**Keywords:** frail, older, disability, cohort study, lifestyle, prevention

## Abstract

There has been less research conducted on the psychological aspects of frailty than on the physical and cognitive characteristics of frailty. Thus, we aimed to define psychological frailty, clarify its prevalence, and investigate the relationship between psychological frailty and lifestyle activity or disability incidence in older adults in Japan. The participants in our study were 4126 older adults (average age 71.7 years) enrolled in the National Center for Geriatrics and Gerontology-i87uStudy of Geriatric Syndromes (NCGG-SGS). We characterized physical frailty of the following as ≥ 3: slow walking speed, muscle weakness, exhaustion, low physical activity, and weight loss. We used the Geriatric Depression Scale 15 items version (GDS-15) to screen for depressive mood, indicated by 5 points or more on the scale. The co-presence of physical frailty and depressive mood was defined as psychological frailty. The incidence of disability was determined using data from the Japanese long-term care insurance system over 49 months. We found that the prevalence of physical frailty, depressive mood, and psychological frailty were 6.9%, 20.3%, and 3.5%, respectively. Logistic regression indicated that the odds ratios for loss of lifestyle activities were significantly higher in participants with psychological frailty for going outdoors using the bus or train, driving a car, using maps to go to unfamiliar places, reading books or newspapers, cognitive stimulation, culture lessons, giving advice, attending community meetings, engaging in hobbies or sports, house cleaning, fieldwork or gardening, and taking care of grandchildren or pets. During the follow-up period, 385 participants (9.3%) developed a disability. The incidence of disability was associated with both physical and psychological frailty in the fully adjusted model. However, no significant association between disability and depressive mood was found. We conclude that individuals with psychological frailty had the highest risk of disability. Future policies should implement disability prevention strategies among older adults with psychological frailty.

## 1. Introduction

Approximately 15% of the world’s population is estimated to live with a disability, and between 110 million and 190 million people aged 15 years or older have significant difficulties in functioning. Moreover, the rates of disability are increasing due to aging populations and an increase in the incidence of chronic health conditions [1].

One of the most problematic expressions of population aging is the clinical condition of frailty. Frailty develops due to age-related decline in many physiological systems that results in vulnerability to sudden health status changes triggered even by minor stressor events [2]. Between 25% to 50% of older adults aged 85 years and over are estimated to have frailty, and thus, have a considerably increased risk of disability, long-term care, falls, and death [3,4]. 

Frailty is a reduced capacity to cope with stressors and a multifaceted geriatric syndrome reflecting multi-system dysfunction. It may involve social, psychological, and emotional aspects, in addition to physical factors [5]. Collard et al. have reported that for individuals aged 65 years or older residing in the community, the prevalence of physical frailty varied significantly from 4% to 59%, and the overall weighted prevalence of frailty was 10.7% [6]. The authors concluded that frailty is common in later life, but different operationalization of the status of frailty has resulted in widely differing prevalence of frailty among the research on the subject [6]. Although there is no universally accepted operational definition of physical frailty, the most commonly used definition of a physical phenotype of frailty was developed by Fried et al. [4].

Alongside physical frailty, depression is one of the most significant health issues among older adults, and its prevalence ranges from 10 to 20% [7]. One meta-analysis found that approximately 4–16% of frail older adults aged 60 and over had serious depression [8], and this percentage increased to 35% in those aged 75 and over [9]. Data from a Singaporean longitudinal aging study revealed that prefrail and frail participants were more likely to show persistent and new depressive symptoms at follow-up [10]. Furthermore, depressive symptoms and antidepressant use were associated with an increased risk of becoming prefrail and frail in a prospective cohort study [11]. In addition, Freiheit et al. found that a frailty index, including physical, cognitive, and psychological factors, is associated with incidences of disability, but the effects of combining physical and psychological aspects on the incidence of disability are not clear [12]. The co-occurrence of frailty and depressive mood and the relationship between this co-occurrence and incidence of disability in the Japanese population has not been established.

In this study, we defined psychological frailty as the co-occurrence of physical frailty and depressive mood to investigate whether psychological frailty affects the incidence of disability in the Japanese population. Furthermore, we also examined the relationship between psychological frailty and activity status, related factors common to frailty and depressive mood. The promotion of protective factors and the reduction of risk factors is essential to the formulation of effective interventions for preventing disability. Prospective observational studies have suggested several common factors related to disability prevention. For example, older persons who had participated to a greater extent in everyday activities were found to have a lower risk of disability and dementia [13,14,15,16,17,18,19,20,21]. These findings suggest that lifestyle activities could be intermediaries between psychological frailty and incidence of disability. Thus, we also examined the relationship between psychological frailty and lifestyle activity, including instrumental activities of daily living (IADL), cognitive activity, social activity, and productive activity. We hypothesized that individuals with psychological frailty have a higher prevalence of loss of lifestyle activities and higher risk of disability incidence compared to those without psychological frailty.

## 2. Experimental Section

### 2.1. Participants

Our national study assessed 5104 individuals aged 65 or older (mean age 71 years) enrolled in the National Center for Geriatrics and Gerontology-Study of Geriatric Syndromes (NCGG-SGS), a Japanese national cohort study [22]. Each participant was recruited from Obu, a residential suburb of Nagoya. The inclusion criteria were being 65 years and over at the time of examination (2011 or 2012) and residing in Obu. Based on previous reports that certain conditions could produce characteristics of disability [23], we excluded participants with a history of depression (*n* = 152), stroke (*n* = 268), Parkinson’s disease (*n* = 16), dementia (*n* = 7), and mini-mental state examination [24] scores < 18 (*n* = 39). We also excluded participants with a functional decline in basic activities of daily living (ADL; *n* = 22), certified long-term care insurance (*n* = 86), and missing data values regarding determinants for frailty, depressive mood, and incidence of disability (*n* = 388). Of the initial 5104 participants, 874 were excluded based on these criteria (Figure 1). Subsequently, the study analyzed data from 4126 older adults (mean age 71.7 ± 5.3 years, 49.2% male). The Ethics Committee of the National Center for Gerontology and Geriatrics approved the study protocol (registered number: 791), and informed consent was obtained from all participants prior to their inclusion.

### 2.2. Operationalization of Psychological Frailty

The assessments were conducted by well-trained assessors with allied health, nursing, or similar qualifications. Before commencement, all of the assessors received training from the authors in the protocols for administering the measures.

The physical frailty phenotype was defined as having limitations in three or more of the following domains: mobility, strength, endurance, physical activity, and nutrition. Walking speed was measured in seconds using a stopwatch, and the participants were asked to walk at a comfortable walking speed. Two markers were used to indicate the start and end of a 2.4-meter path, and a 2-meter section had to be traversed before the starting marker so that the participants would already be walking at a comfortable pace. The participants were asked to continue walking for an additional two meters afterward to ensure a consistent walking pace while on the timed section, and low mobility was established as <1.0 m/s [25,26]. Grip strength was measured using a Smedley-type handheld dynamometer (GRIP-D; Takei Ltd., Niigata, Japan), and low grip strength was established according to a sex-specific cutoff (male: < 26 kg, female: < 17 kg) [27]. Exhaustion was considered to be present if the participant responded “yes” to the following question included on the Kihon Checklist [28], a self-reported comprehensive health checklist developed by the Japanese Ministry of Health, Labour, and Welfare: “In the last two weeks, have you felt tired without a reason?” Physical activity was evaluated with the following questions: (1) “Do you engage in moderate levels of physical exercise or sports for your health?”; and (2) “Do you engage in low levels of physical exercise for your health?” If the participants answered “no” to both questions, we considered that they engaged in low levels of activity [25]. Weight loss was assessed from the response to the question: “Have you lost 2 kg or more in the past six months?” [28].

We used the Geriatric Depression Scale 15 items version (GDS-15) to assess depressive mood [29]. The GDS-15 has been widely recommended as a brief screening measurement for late-life depression [30] and has been useful in detecting late-life major depression in primary care settings [31]. Many studies have determined the validity [30,32] and internal consistency reliability [32] of the GDS-15. We selected the cutoff point of 4/5 in the GDS-15 because, for screening purposes, cutoff points that yield high levels of sensitivity and negative predictive value are preferred, and a previous study revealed that use of the cutoff point of 4/5 (non-case/case) for the GDS-15 produced robust results [33]. 

The participants were divided into the following four categories: (1) robust older adults with no physical frailty or depressive mood (robust group); (2) physically frail older adults without depressive moods (physical frailty group); (3) non-physically frail older adults with depressive moods (depression group); and (4) physically frail older adults with depressive moods (psychological frailty group).

### 2.3. Determination of Disability

The participants were tracked monthly for Japanese public long-term care insurance (LTCI) certification, as recorded by the Japanese LTCI system managed by each municipal government. The certification for LTCI has been reported in detail elsewhere [34]. Briefly, all individuals aged 65 years or older, or those aged between 40–64 years who suffer from age-related diseases, are eligible for LTCI benefits in Japan. When a person applies to their municipality for LTCI benefits, an authorized care manager examines their physical and mental status using a standardized questionnaire. Then the certification board, which includes medical doctors and nurses, determines the level of long-term care they require based on the estimated time required for care, as well as on comments from the applicant’s family physician. The LTCI classifies a person as “Support Level 1 or 2” to indicate the need for assistance with basic ADL or “Care Level 1 through 5” to indicate the need for continuous care [34]. In this study, disability was defined as an LTCI certification of any level, and we defined disability onset as the point at which a participant was certified by the LTCI to require support or care.

### 2.4. Measurements of Lifestyle Activity

The participants completed a questionnaire comprising 15 questions regarding IADL, cognitive activity, social activity, and productive activity as different elements of lifestyle activity [35]. The following questions determined IADL activity: (1) “Do you go outdoors using the bus or train?”; (2) “Do you engage in cash handling and banking?”; (3) “Do you drive a car?”; and (4) “Do you use maps to go to unfamiliar places?” The following items determined cognitive activity: (5) “Do you read books or newspapers?”; (6) “Do you engage in cognitive stimulation such as board games and learning?”; (7) “Do you engage in cultural classes?”; and (8) “Do you use a personal computer?” The following questions measured social activity: (9) “Are you sometimes called on for advice?”; (10) “Do you attend meetings in the community?”; and (11) “Do you engage in hobbies or sports activities?” Finally, the following items determined productive activity: (12) “Do you engage in housecleaning?”; (13) “Do you engage in fieldwork or gardening?”; (14) “Do you take care of grandchildren or pets?”; and (15) “Do you engage in paid work?” Answers of “yes” were determined to be positive responses.

### 2.5. Potential Confounding Factors

We selected six demographic variables and five primary diseases or a geriatric syndrome as the possible confounding factors of ADL decline (Table 1) [36,37,38]. The demographic variables included age, sex, education, medication, current smoking, and living alone. Medical information was obtained via self-reporting and interview surveys. The following were reported: heart disease, pulmonary disease, hypertension, diabetes, osteoarthritis, and history of falls.

### 2.6. Statistical Analysis

The prevalence of physical frailty, depressive mood, and psychological frailty were explored, and we compared age- and sex-specific prevalence rates of psychological frailty using Chi-square tests. Baseline characteristics were compared according to frailty status and disability using t-tests, one-way analysis of variance, and Chi-square tests. To identify the impact of psychological frailty on the incidence of disability using Chi-square tests, we used adjusted standardized residuals. The adjusted standardized residuals followed the t distribution, with > 1.96, *p* < 0.05 and > 2.56, *p* < 0.01. Multiple logistic regression models were used to analyze the associations between lifestyle activity status and psychological frailty adjusted for potential confounding factors. Adjusted odds ratios (ORs) and their 95% confidence intervals (95% CIs) were estimated, and we calculated the cumulative incidence of disability during follow-up. Cox proportional hazards regression models were used to determine the associations between cognitive impairment and the incidence of disability. Model 1 was crude, and Model 2 was adjusted for potential confounding factors. We estimated adjusted hazard ratios (HRs) and 95% CIs for the incidence of disability. All of the data management and statistical analyses were performed using the IBM SPSS Statistics 24.0 software package (IBM Tokyo, Japan).

## 3. Results

The NCGG-SGS identified 285 (6.9%) older participants with symptoms of physical frailty, 836 (20.3%) with a depressive mood, and 146 (3.5%) who had psychological frailty, defined as the combination of physical frailty and depressive mood (Figure 1). In the residual analyses, the physical frailty and psychological frailty groups included significantly more participants with incidents of disability (*p* < 0.01). We found that the prevalence of physical frailty, depressive mood, and psychological frailty increased with age (*p* < 0.01) (Figure 2). The prevalence of physical frailty was higher in female participants (*p* < 0.05), but there were no significant sex-specific differences in the prevalence of depressive mood or psychological frailty (Figure 2).

Table 1 presents the possible confounding factors for disability incidence among the participants, grouped according to frailty status and the presence of a disability. Significant differences between the frailty subgroups were found for age, educational level, medication, living alone, heart disease, pulmonary disease, hypertension, diabetes, osteoarthritis, fall history, and all lifestyle activities (Table 1). Additionally, significant differences between the participants according to the presence of disability were found for age, sex, educational level, medication, current smoking, living alone, heart disease, hypertension, diabetes, osteoarthritis, fall history, and all but three lifestyle activities (Table 1).

We found several significant relationships between psychological frailty and lifestyle activities. Individuals with psychological frailty had higher ORs compared with robust group for not engaging in the following activities: going out using the bus or train, driving a car, using maps to go to unfamiliar places, reading books and newspapers, cognitive stimulation, culture lessons, giving advice, attending community meetings, engaging in hobbies and sports, house cleaning, fieldwork or gardening, and taking care of grandchildren or pets (Table 2).

From the results, 385 participants (9.3%) had an incident of disability, 39 (0.9%) moved away from Obu, and 78 (1.9%) died during the follow-up period (average 49.2 ± 9.4 months). The disability incidence in the robust, physical frailty, depressive mood, and psychological frailty groups was 6.7%, 30.9%, 10.9%, and 38.4%, respectively (Figure 3). Cox proportional hazards regression models were used to analyze the associations between frailty and disability incidence (Table 3, Figure 3). In the crude model (Model 1), a significantly higher disability incidence was found for the physical frailty, depressive mood, and psychological frailty groups. Different HRs were obtained in the fully adjusted model (Model 2), and no significant association was found between depressive mood and disability incidence. In Model 2, age, education level, medication, fall history, not participating in culture lesson, and not engaging in fieldwork or gardening correlated positively with an incident of disability.

## 4. Discussion

This study presents original data regarding vulnerability to physical and psychological decline among 4126 older community dwellers. We revealed that the incidence of disability was associated with both physical and psychological frailty and the individuals with psychological frailty had the highest risk of disability.

A growing body of evidence has indicated a connection between physical frailty and depressive mood. Several studies have reported a longitudinal association between frailty and rate of depressive mood in older community-dwelling individuals. A systematic review evaluated the co-occurrence of physical frailty and depression in adults aged 60 and older and found that the prevalence of frailty was around 4–16% [8]. The NCGG-SGS identified 6.9% older participants with physical frailty, 20.3% with a depressive mood, and 3.5% with psychological frailty, defined as the combination of physical frailty and depressive mood. Our participants demonstrated a lower prevalence of the co-occurrence of physical frailty and depression compared to previous studies [8]. Although the prevalence of frailty has been reported to range between 4% to 59% in community studies [6], there is a marked variation among these studies in terms of geographic difference. For example, a recent systematic review identified that the global prevalence of frailty as defined by Fried et al.’s criteria varied from 3.9% (China) to 26.0% (India) in low-income and middle-income countries [39]. The pooled prevalence of frailty was 11.1% (95% CI, (8.9, 13.4), I2 = 91.4%, *p* < 0.001) in men and 15.2% (95% CI (12.5, 18.1), I2 = 95.2%, *p* < 0.001) in women [39]. In European studies, the prevalence rates in community settings varied, ranging from 2% to 60% with a median rate of 10.8% [40]. The meta-analysis of the prevalence of frailty in community-dwelling Japanese older adults revealed that the pooled prevalence was 7.4% (95% CI (6.1, 9.0)) [41]. The participants in this study had a relatively lower prevalence rate (6.9%) of physical frailty compared to previous studies [39,40,41], which may have, in turn, affected the lower prevalence of psychological frailty. Our participants were not recruited randomly, and we excluded those with a history of depression, stroke, Parkinson’s disease, dementia, low mini-mental state examination scores, basic ADL decline, and certified long-term care insurance. This may lead to an underestimation of the prevalence of frailty.

Regarding the measurement methods, the principal difference between Fried’s frailty criteria and the NCGG-SGS is the cut-off point for walking speed: Fried’s criteria it is set at 0.65 m/s (height ≤ 173 cm), whereas in the NCGG-SGS it is 1.0 m/s. Walking speed has been found to be a strong predictor of adverse events such as disability [42,43,44,45,46,47,48], mortality [43,44,49,50], hospitalization [43,44,46,51], and falls [51,52]. The cut-off point for walking speed in the present study was 1.0 m/s, which appears to be a critical point for predicting future functional decline among community-dwelling older individuals [43,44,46,47,48]. These results suggest that walking speed may be the most useful measurement for determining frailty [53] and predicting future functional decline in older adults. The low prevalence of physical frailty despite the higher cut-off point for walking speed in this study may be due to the participants’ better health status compared to the participants in previous studies.

Our logistic model revealed several significant relationships between psychological frailty and lifestyle activities, which included going out using the bus or train, driving a car, using maps to go to unfamiliar places, reading books and newspapers, cognitive stimulation, culture lessons, giving advice, attending community meetings, engaging in hobbies and sports, house cleaning, fieldwork or gardening, and taking care of grandchildren or pets. Although causality cannot be inferred in this cross-sectional analysis, psychological frailty does have a significant relationship with specific psychosocial activities, which may be helpful when considering preventive interventions. Previous studies identified that the probability of dementia was significantly lower in participants who engaged in daily conversation, drove a car, went shopping, and did fieldwork or gardening [54], and the probability of reversion from mild cognitive impairment to normal cognition was significantly higher in participants who engaged in driving a car, using maps to travel to unfamiliar places, reading books or newspapers, cultural classes, meetings in the community, hobbies or sports activities, and fieldwork or gardening [35]. Risk factors common to psychological frailty and the onset of dementia suggest an association between the two. As such, the implementation of these specific activities may be effective in preventing the development of psychological frailty and dementia. For instance, driving cessation is associated with several negative consequences, such as declined general health [55], cognitive decline [56], depressive symptoms [57], increased risk for long-term care institutionalization [58], and mortality [59]. The results suggest that driving is associated with maintenance of cognitive function. We revealed that a driving skill program improved safe driving performance significantly in older adults with cognitive impairments who had potentially high risk of a car accident [60]. We demonstrated that it was necessary to examine effective intervention strategies for older adults to prevent psychological frailty.

The Cox proportional hazards regression models revealed that participants with psychological frailty demonstrated significantly higher disability incidence compared to participants without physical frailty or depressive moods. The data from our cohort study demonstrate that physical frailty combined with depression symptoms increases the risk of disability incidence more than physical frailty alone. Thus, healthcare providers are advised to perform both physical and psychological assessments to evaluate disability risk and deliver healthcare services targeted at high-risk individuals, especially among psychologically frail older persons.

An important limitation of our study is that the participants were not recruited randomly for the NCGG-SGS. This may lead to an underestimation of the prevalence of frailty and depressive moods, as the participants were relatively healthy older persons who were able to access health checkups from their homes. Second, we were not able to contact informants, such as family members, to verify medical records, lifestyle information, and asymptomatic aberrant behavior. Despite these limitations, one notable strength of the present study is the size of the cohort assessed in a specific community and the fact that our findings are backed by comprehensive geriatric assessments intended to identify frailty, depressive mood, and specific activities.

## 5. Conclusions

We defined psychological frailty as the co-presence of physical frailty and depressive mood. The prevalence of psychological frailty was 3.5% among the older adults who participated in this study, and individuals with psychological frailty had a decreased level of lifestyle activities, such as IADL and cognitive or social activities. The incidence of disability was associated with psychological frailty in the fully adjusted model; therefore, we demonstrated that psychological frailty could be used in gerontology and geriatrics as an indicator of disability prevention in community-living older adults. Further research is required to examine whether the results are common among hospitalized patients or whether the results are the same in countries other than Japan. As a reduction in lifestyle activities such as going out was associated with psychological frailty, it may be best to promote these activities as part of prevention strategies.

## Figures and Tables

**Figure 1 jcm-08-01554-f001:**
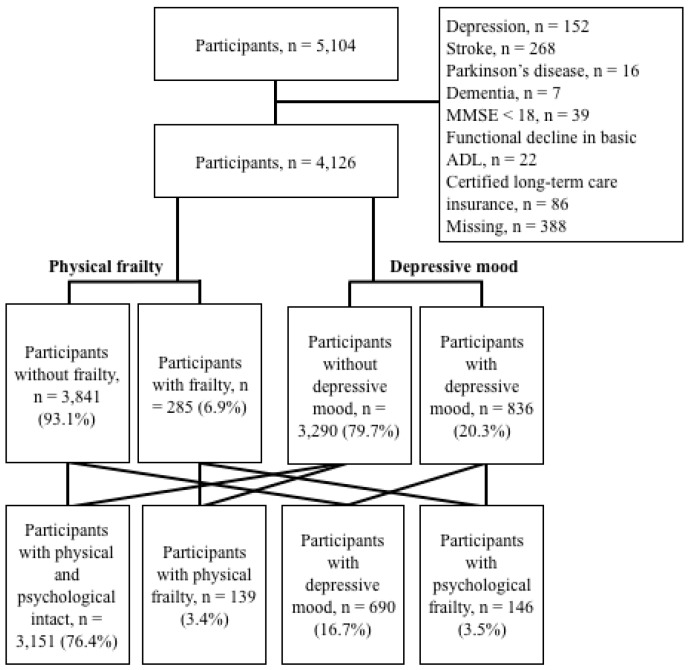
Participants’ flow and frailty status.

**Figure 2 jcm-08-01554-f002:**
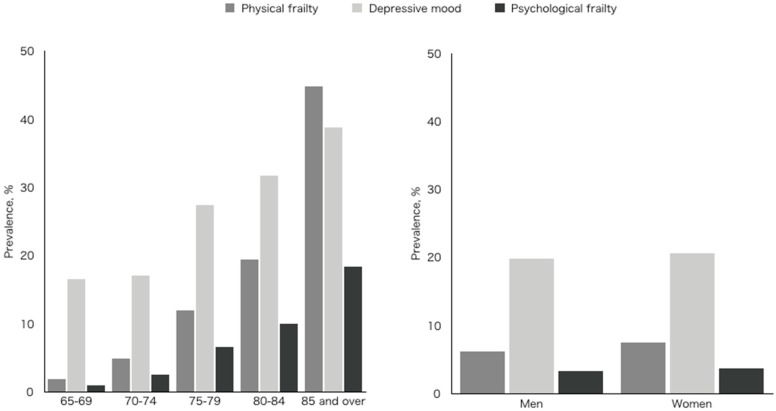
Prevalence of psychological frailty by age and sex.

**Figure 3 jcm-08-01554-f003:**
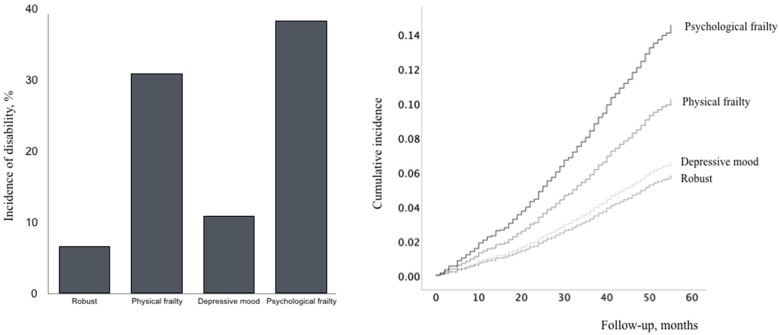
Cox proportional hazards regression models of disability incidence according to frailty status.

**Table 1 jcm-08-01554-t001:** Comparisons of baseline characteristics according to frailty status and between participants with and without disability.

	Between the Frailty Status					Between the Participants with and without Disability Incidence		
	Group *R* (*n* = 3151)	Group PhF (*n* = 139)	Group D (*n* = 690)	Group PF (*n* = 146)	*p* Value	Participants with Dementia (*n* = 385)	Participants without Dementia (*n* = 3741)	*p* Value
**Demographic variables**								
Age, years	71.1 ± 4.8	77.7 ± 6.6	72.4 ± 5.7	77.1 ± 6.3	<0.001	77.7 ± 6.2	71.1 ± 4.9	<0.001
Sex, male	1567 (49.7)	59 (42.4)	338 (49.0)	66 (45.2)	0.280	1880 (50.3)	150 (39.0)	<0.001
Education, years	11.6 ± 2.5	10.4 ± 2.7	11.0 ± 2.4	10.3 ± 2.5	<0.001	10.2 ± 2.5	11.6 ± 2.5	<0.001
Medication, *n*	1.8 ± 1.9	2.7 ± 2.5	2.2 ± 2.2	2.9 ± 2.6	<0.001	2.8 ± 2.5	1.8 ± 1.9	<0.001
Smoking, yes	305 (9.7)	16 (11.5)	74 (10.7)	11 (7.5)	0.570	380 (10.2)	26 (6.8)	0.033
Living alone, yes	261 (8.3)	19 (13.7)	97 (14.1)	13 (8.9)	<0.001	330 (8.8)	60 (15.6)	<0.001
**Primary diseases or geriatric syndromes**								
Heart disease, yes	471 (14.9)	25 (18.0)	113 (16.4)	40 (27.4)	0.001	571 (15.3)	78 (20.3)	0.010
Pulmonary disease, yes	317 (10.1)	20 (14.4)	90 (13.0)	21 (14.4)	0.027	401 (10.7)	47 (12.2)	0.371
Hypertension, yes	1353 (42.9)	71 (51.1)	316 (45.8)	79 (54.1)	0.011	1616 (43.2)	203 (52.7)	<0.001
Diabetes, yes	395 (12.5)	28 (20.1)	90 (13.0)	37 (25.3)	<0.001	478 (12.8)	72 (18.7)	0.001
Osteoarthritis, yes	389 (12.3)	25 (18.0)	117 (17.0)	27 (18.5)	0.001	489 (13.1)	69 (17.9)	0.008
Fall history, yes	365 (11.6)	35 (25.2)	141 (20.4)	41 (28.1)	<0.001	488 (13.0)	94 (24.4)	<0.001
**Lifestyle activity**								
Going out using the bus or train, no	224 (7.1)	27 (19.4)	91 (13.2)	35 (24.0)	<0.001	316 (8.4)	61 (15.8)	<0.001
Cash handling and banking, no	289 (9.2)	23 (16.5)	87 (12.6)	18 (12.3)	0.002	377 (10.1)	40 (10.4)	0.847
Driving a car, no	757 (24.0)	78 (56.1)	244 (35.4)	76 (52.1)	<0.001	950 (25.4)	205 (53.2)	<0.001
Using maps to go to unfamiliar places, no	1029 (32.7)	81 (58.3)	318 (46.1)	89 (61.0)	<0.001	1299 (34.7)	218 (56.6)	<0.001
Reading books or newspapers, no	101 (3.2)	10 (7.2)	33 (4.8)	18 (12.3)	<0.001	137 (3.7)	25 (6.5)	0.006
Cognitive stimulation such as board games and learning, no	1385 (44.0)	94 (67.6)	481 (69.7)	113 (77.4)	<0.001	1828 (48.9)	245 (63.6)	<0.001
Culture lesson, no	1690 (53.6)	104 (74.8)	493 (71.4)	124 (84.9)	<0.001	2178 (58.2)	233 (60.5)	0.383
Using personal computer, no	1939 (61.5)	109 (78.4)	526 (76.2)	123 (84.2)	<0.001	2370 (63.4)	327 (84.9)	<0.001
Giving advice, no	170 (5.4)	11 (7.9)	103 (14.9)	43 (29.5)	<0.001	265 (7.1)	62 (16.1)	<0.001
Attending meetings in the community, no	1381 (43.8)	80 (57.6)	423 (61.3)	114 (78.1)	<0.001	1802 (48.2)	196 (50.9)	0.306
Engaging in hobbies or sports activities, no	602 (19.1)	70 (50.4)	294 (42.6)	101 (69.2)	<0.001	906 (24.2)	161 (41.8)	<0.001
House cleaning, no	365 (11.6)	28 (20.1)	89 (12.9)	35 (24.0)	<0.001	454 (12.1)	63 (16.4)	0.017
Fieldwork or gardening, no	760 (24.1)	41 (29.5)	258 (37.4)	71 (48.6)	<0.001	995 (26.6)	135 (35.1)	<0.001
Taking care of grandchildren or pets, no	1312 (41.6)	68 (48.9)	370 (53.6)	93 (63.7)	<0.001	1621 (43.3)	222 (57.7)	<0.001
Paid work, no	2109 (66.9)	107 (77.0)	530 (76.8)	120 (82.2)	<0.001	2550 (68.2)	316 (82.1)	<0.001

Group R: Robust group, Group PhF: Physical frailty group, Group D: Depressive mood group, Group PF: Psychological frailty group.

**Table 2 jcm-08-01554-t002:** Relationships between frailty status and lifestyle activity.

		Odds Ratio (95% CI)	*p* Value
Going out using the bus or train	Physical frailty	3.65 (2.27–5.87)	<0.001
	Depressive mood	1.88 (1.44–2.45)	<0.001
	Psychological frailty	4.63 (2.99–7.16)	<0.001
Cash handling and banking, yes	Physical frailty	2.55 (1.51–4.31)	<0.001
	Depressive mood	1.42 (1.08–1.86)	0.012
	Psychological frailty	1.51 (0.86–2.65)	0.150
Driving a car, yes	Physical frailty	2.48 (1.58–3.88)	<0.001
	Depressive mood	1.72 (1.38–2.14)	<0.001
	Psychological frailty	2.24 (1.46–3.46)	<0.001
Using maps to go to unfamiliar places, yes	Physical frailty	2.04 (1.39–3.00)	<0.001
	Depressive mood	1.64 (1.36–1.97)	<0.001
	Psychological frailty	2.48 (1.69–3.64)	<0.001
Reading books and newspapers, yes	Physical frailty	1.97 (0.96–4.06)	0.066
	Depressive mood	1.31 (0.87–1.99)	0.199
	Psychological frailty	3.82 (2.12–6.91)	<0.001
Cognitive stimulation such as board games and learning, yes	Physical frailty	2.25 (1.53–3.32)	<0.001
	Depressive mood	2.77 (2.30–3.33)	<0.001
	Psychological frailty	3.84 (2.53–5.83)	<0.001
Culture lesson, yes	Physical frailty	3.19 (2.10–4.83)	<0.001
	Depressive mood	2.24 (1.85–2.72)	<0.001
	Psychological frailty	5.87 (3.63–9.49)	<0.001
Using a personal computer, yes	Physical frailty	0.94 (0.58–1.52)	0.804
	Depressive mood	1.79 (1.45–2.23)	<0.001
	Psychological frailty	1.64 (0.98–2.75)	0.061
Giving advice, yes	Physical frailty	1.25 (0.65–2.41)	0.512
	Depressive mood	2.83 (2.16–3.69)	<0.001
	Psychological frailty	6.05 (3.97–9.22)	<0.001
Attending meetings in the community, yes	Physical frailty	2.11 (1.47–3.02)	<0.001
	Depressive mood	2.14 (1.80–2.54)	<0.001
	Psychological frailty	5.70 (3.77–8.61)	<0.001
Engaging in hobbies and sports activities, yes	Physical frailty	3.40 (2.37–4.90)	<0.001
	Depressive mood	2.90 (2.42–3.47)	<0.001
	Psychological frailty	8.09 (5.53–11.85)	<0.001
House cleaning, yes	Physical frailty	2.36 (1.42–3.91)	0.001
	Depressive mood	1.22 (0.94–1.60)	0.142
	Psychological frailty	3.08 (1.93–4.91)	<0.001
Fieldwork or gardening, yes	Physical frailty	1.37 (0.93–2.03)	0.112
	Depressive mood	1.87 (1.56–2.23)	<0.001
	Psychological frailty	3.22 (2.27–4.58)	<0.001
Taking care of grandchildren or pets, yes	Physical frailty	0.98 (0.69–1.41)	0.922
	Depressive mood	1.52 (1.28–1.80)	<0.001
	Psychological frailty	1.97 (1.38–2.82)	<0.001
Paid work, yes	Physical frailty	0.74 (0.48–1.15)	0.179
	Depressive mood	1.55 (1.27–1.90)	<0.001
	Psychological frailty	1.17 (0.74–1.85)	0.514

**Table 3 jcm-08-01554-t003:** Hazard ratios for disability according to frailty status and confounding factors.

	Model 1		Model 2	
	Hazard Ratio (95% CI)	*p* Value	Hazard Ratio (95% CI)	*p* Value
**Demographic variables**				
Age, years			1.14 (1.11–1.16)	<0.001
Sex, male			1.19 (0.89–1.60)	0.233
Education, years			0.93 (0.89–0.98)	0.004
Medication, *n*			1.08 (1.03–1.13)	0.002
Smoking, yes			1.26 (0.82–1.91)	0.289
Living alone, yes			1.08 (0.81–1.44)	0.613
**Primary diseases or geriatric syndromes**				
Heart disease, yes			1.03 (0.79–1.33)	0.854
Pulmonary disease, yes			0.98 (0.72–1.34)	0.918
Hypertension, yes			1.05 (0.85–1.30)	0.647
Diabetes, yes			1.21 (0.92–1.60)	0.172
Osteoarthritis, yes			0.90 (0.69–1.18)	0.448
Fall history, yes			1.34 (1.05–1.72)	0.019
**Lifestyle activity**				
Going out using the bus or train, no			1.22 (0.89–1.67)	0.222
Cash handling and banking, no			0.94 (0.65–1.36)	0.730
Driving a car, no			1.23 (0.95–1.61)	0.122
Using maps to go to unfamiliar places, no			1.18 (0.93–1.49)	0.179
Reading books or newspapers, no			1.03 (0.67–1.58)	0.894
Cognitive stimulation such as board games and learning, no			1.05 (0.82–1.34)	0.715
Culture lesson, no			0.73 (0.56–0.95)	0.018
Using a personal computer, no			1.21 (0.88–1.66)	0.253
Giving advice, no			1.20 (0.88–1.63)	0.261
Attending meetings in the community, no			0.80 (0.64–1.01)	0.058
Engaging in hobbies or sports activities, no			1.26 (0.97–1.64)	0.080
House cleaning, no			1.12 (0.82–1.52)	0.488
Fieldwork or gardening, no			1.28 (1.02–1.60)	0.033
Taking care of grandchildren or pets, no			1.07 (0.86–1.32)	0.573
Paid work, no			0.96 (0.72–1.27)	0.753
**Frailty status**				
Robust	1		1	
Physical frailty	5.66 (4.08–7.86)	<0.001	1.69 (1.16–2.46)	0.006
Depressive mood	1.67 (1.28–2.17)	<0.001	1.05 (0.79–1.39)	0.734
Psychological frailty	7.67 (5.71–10.30)	<0.001	2.24 (1.57–3.20)	<0.001

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
