# Peer review of "Prevalence of Psychological Frailty in Japan: NCGG-SGS as a Japanese National Cohort Study"

_jcm, 2019, doi:10.3390/jcm8101554_

Round 1

Reviewer 1 Report

The paper defines “psychological fragility” as “a  connection between physical frailty and depressive mood”. However, psychological fragility in aging can be due not only to depressive mood but also to psychological variables such as anxiety, self-efficacy or erroneous beliefs associated with age. Therefore, instead of “psychological fragility” it would be more adequate for the paper to refer to “a connection between physical frailty and depressive mood”. 

On the other hand, I would like to know why the authors have not used standardized scales to measure IADL, such as “The Lawton and Brody index”. For the particular study presented in the paper, the autonomy sub-scale of the “WHOWOL-OLD” seems suited. Notice that any of both scales would facilitate the replication of this study.

Author Response

Comments

The paper defines “psychological fragility” as “a connection between physical frailty and depressive mood.” However, psychological fragility in aging can be due not only to depressive mood but also to psychological variables such as anxiety, self-efficacy or erroneous beliefs associated with age. Therefore, instead of “psychological fragility” it would be more adequate for the paper to refer to “a connection between physical frailty and depressive mood.” 

Response

Thank you for your critical comments. I agree with your comment that psychological fragility in aging can be due not only to depressive mood but also to psychological variables. Unfortunately, our database did not include the measurements of anxiety, self-efficacy, or erroneous beliefs. The GDS‐15 has been widely recommended as a brief screening instrument for late‐life depression and has been found to be useful in detecting late‐life major depression in primary care settings. Different factorial structure models of the GDS were proposed in different samples. In a nonclinical sample of older adults, while Sheikh et al. found a five-factor solution, which accounted for 42% of the total variance (“Sad mood,” “Lack of energy,” “Positive mood,” “Agitation,” and “Social withdrawal”),1 Parmelee et al. found a six-factor solution, accounting for 52.3% of the total variance (“General dysphoria,” “Worry,” “Withdrawal/apathy,” “Vigor,” “Decreased concentration,” and “Anxiety”).2 We have selected GDS, which includes these various components and can be easily measured, as a screening test for the psychological state of older adults.

Sheikh JI, et al. Proposed factor structure of the Geriatric Depression Scale. Int Psychogeriatr. 3(1): 23-28, 1991. Parmelee PA, et al. Psychometric properties of the Geriatric Depression Scale among the institutionalized aged: Psychol Assess. 1(4): 331, 1989.

Comments

On the other hand, I would like to know why the authors have not used standardized scales to measure IADL, such as “The Lawton and Brody index.” For the particular study presented in the paper, the autonomy sub-scale of the “WHOWOL-OLD” seems suitable. Notice that any of both scales would facilitate the replication of this study.

Response

Thank you for your advice on the outcomes. In this study, we used data from the long-term care insurance (LTCI) to which all older people belong, to prevent data loss of the primary outcome. Japan implemented the LTCI system on April 1, 2000, and every Japanese person aged 65 and older is eligible for benefits based strictly on their physical and mental frailty or disability.3 The LTCI is a mandatory national insurance plan for older people in Japan, and it was instituted to support the independence of older people, rather than simply providing personal care. The certification for LTCI has been reported in detail elsewhere.3 Briefly, all individuals aged 65 years or older, or those aged 40–64 years who suffer from age-related diseases are eligible for LTCI benefits in Japan. When a person applies to their municipality for LTCI benefits, an authorized care manager examines their physical and mental status using a standardized questionnaire. Then the certification board, which includes medical doctors and nurses, determines the level of long-term care necessary based on the estimated time required for care, as well as on comments from the applicant’s family physician. The LTCI certifications consist of the following eight levels: independent, support levels 1–2, and care levels 1–5. We defined a person certified as LTCI support level 1 as a person who can independently carry out basic activities of daily living but requires some assistance in instrumental activities of daily living, or higher as a disability. We added an explanation about LTCI to the text.

Tsutsui T, et al. Care-needs certification in the long-term care insurance system of Japan. J Am Geriatr Soc. 53(3): 522-527, 2005.

Reviewer 2 Report

General comment:

The authors researched the prevalence of psychological frailty in elderly people. The aim of the authors was to define psychological frailty, clarify its prevalence, and investigate the relationship between psychological frailty and lifestyle activity or disability incidence in older adults.

The results showed that the prevalence of physical frailty, depressive mood, and psychological frailty was 6.9%, 20.3%, and 3.5%, respectively. Individuals with psychological frailty had the highest risk of disability.

Strengths:

The great effort was to investigate and follow-up a large group. The participants were 4,126 older adults enrolled in the National Center for Geriatrics and Gerontology Study of Geriatric Syndromes. The incidence of disability was determined using data from the Japanese long-term care insurance system over 49 months.

This study by Shimada et al. is very remarkable. The psychological component of the frailty was neglected, and therefore I would like to commend the authors of the study. The advantage is the relatively large number of participants.  

Weaknesses:

Despite the solid study design, results may be underestimated due to lack of randomization and absence of detailed clinical data on subjects. However, the authors are aware of these weaknesses and have taken them into account in study limitations. Also, multiple investigators may have influenced differences in the uniformity of the study protocol. Nevertheless, authors state that the assessments were conducted by well-trained assessors with nursing, allied health, or similar qualifications, thus limiting the risk of misconduct.

There are several comments which should be addressed.

Title:

Authors should consider rewriting the title.

The title properly reflects the subject of the paper, but it should be considered whether to change it to “Prevalence of psychological frailty in the older adults in Japan”. Local groups can substantially differ because of the large variations across countries.

Moreover, the prevalence of frailty is highly dependent on the measurement instrument used. Collard et al. report that in community-dwelling individuals aged 65 years or older, the prevalence of frailty varies significantly from 4.0% to 59.1% [Collard RM, Boter H, Schoevers RA, Oude Voshaar RC. Prevalence of frailty in community-dwelling older persons: a systematic review. J Am Geriatr Soc. 2012;60:1487–92].

Frailty is common in later life, but different operationalization of frailty status causes significant differences in prevalence between studies.

Introduction:

The introduction is clear, short and simple. It sets out and justifies the aim of the study. The literature review includes other research, but they are not enough. The authors should consider some recent publications about frailty and their risk factors. Only a few papers have been published in the last five years and recently, there have been a lot of new studies on frailty.

Experimental section:

If applicable, please provide the number of study protocol registered with the Bioethical Committee.

As to the inclusion and exclusion criteria, what about patients with chronic kidney disease? Were they in the study group?

In end-stage renal disease, uremia symptoms are often accompanied by frailty. These symptoms are linked to biological aging and occur at an earlier age in chronic dialysis patients than in the general population, which strongly correlates with prognosis. For this reason, ESRD can be treated as a model of accelerated aging. There are many similarities between uremia stages and aging, so the obtained results from hemodialysis patients can be extrapolated to the population of older individuals.

A flow chart with recruitment of participants, drop-outs (deaths and others) with the final number of participants analyzed would facilitate the readability of the manuscript.

Were the known Katz Index [Katz S., Ford A.B., Moskowitz R.W., Jackson B.A., Jaffe M.W. Studies of Illness in the Aged. The Index of ADL: A Standardized Measure of Biological and Psychosocial Function. JAMA. 1963;185:914–919] and Lawton IADL Index [Lawton M.P., Brody E.M. Assessment of older people: Self-maintaining and Instrumental Activities of Daily Living. Gerontologist. 1969;9:179–186] used, their modifications or some other scales?

The system of classification is not transparent to the public. Authors need to be clearer. What are the mentioned LTCI (L. 122) and ADL or “Care Level 1 through 5” (L. 124) support scales? Would it be possible to provide them as supplementary data? Adding background information about LTCI and ADL should be considered as well.

It is unclear which scale for the assessment of functional status was used. The subsection Measurements of life style activity (L. 127–139) is also unclear.

The citation 33 refers to the study of Shimada et al., which includes a scale that contains 16 questions. The scale used in this study consists of 15 questions. Why was the question “Do you talk with other people every day?” not included in the present study?

Describe what was encompassed in the geriatric assessment. Was it a comprehensive geriatric assessment?

The methods lack clarity and are in some parts confusing.

Results:

The results are almost clearly formatted and presented. The units and other notations are correct. Graphs, axis heading, and data labels are readable. Table 1: The groups are defined in the headline of the table. Adjustment this long table for publication would certainly make it easier to read.

A good idea would be to use abbreviations and indicate on the bottom of the table: group R; group PhF, group D, group PF.

Discussion:

Like in the introduction, some recent publications should be integrated into the discussion section.

Similar to the organizing info issue in the intro, list info about the previous findings in relations to the main findings of the present study.

Organization of information regarding previous literature and the data interpretation especially clinical relevance of the findings could be improved. In literature prevalence of frailty is found higher than prevalence in the present study. Please try to explain it further. In such an important issue for geriatric medicine, the discussion is too short; please go a bit deeper with some more relevant reference. As the authors themselves note the level of independence in activities of daily living rather than life expectancy are of the utmost importance at this age.

There is an increasingly wide literature about frailty worldwide. These issues should be more fully debated in the manuscript noting that the findings in this study represent a locality-based study which may/may not be replicated elsewhere. Substantial differences in the prevalence of frailty across countries are observed, with a steep increase with aging.

Conclusions:

The conclusions were correctly presented

The observations of Shimada et al. confirm the multidimensional scope of frailty: a syndrome that includes both physical and psychological conditions.

Conclusions support the concept of a psychological dimension of physical frailty in geriatric inpatients.

The prognostic value of an assessment that accounts for both physical and mental components of frailty is superior to the separate values of these components.

As mentioned above, local groups can vary widely. Thus, conclusions may not necessarily have universal range and need further studies.

Minor issues:

L. 78: The phrase ‘We excluded...’ should be capitalized

L. 121: It should read ‘In this study’

Author Response

General comment:

The authors researched the prevalence of psychological frailty in elderly people. The aim of the authors was to define psychological frailty, clarify its prevalence, and investigate the relationship between psychological frailty and lifestyle activity or disability incidence in older adults.

The results showed that the prevalence of physical frailty, depressive mood, and psychological frailty was 6.9%, 20.3%, and 3.5%, respectively. Individuals with psychological frailty had the highest risk of disability.

Strengths:

The great effort was to investigate and follow-up a large group. The participants were 4,126 older adults enrolled in the National Center for Geriatrics and Gerontology Study of Geriatric Syndromes. The incidence of disability was determined using data from the Japanese long-term care insurance system over 49 months.

This study by Shimada et al. is very remarkable. The psychological component of the frailty was neglected, and therefore I would like to commend the authors of the study. The advantage is the relatively large number of participants.  

Weaknesses:

Despite the solid study design, results may be underestimated due to lack of randomization and absence of detailed clinical data on subjects. However, the authors are aware of these weaknesses and have taken them into account in study limitations. Also, multiple investigators may have influenced differences in the uniformity of the study protocol. Nevertheless, authors state that the assessments were conducted by well-trained assessors with nursing, allied health, or similar qualifications, thus limiting the risk of misconduct.

Response

Thank you for your detailed review. We carefully reviewed your valuable comments and revised the manuscript.

Comments

There are several comments which should be addressed.

Title:

Authors should consider rewriting the title.

The title properly reflects the subject of the paper, but it should be considered whether to change it to “Prevalence of psychological frailty in the older adults in Japan.” Local groups can substantially differ because of the large variations across countries.

Moreover, the prevalence of frailty is highly dependent on the measurement instrument used. Collard et al. report that in community-dwelling individuals aged 65 years or older, the prevalence of frailty varies significantly from 4.0% to 59.1% [Collard RM, Boter H, Schoevers RA, Oude Voshaar RC. Prevalence of frailty in community-dwelling older persons: a systematic review. J Am Geriatr Soc. 2012;60:1487–92].

Frailty is common in later life, but different operationalization of frailty status causes significant differences in prevalence between studies.

Response

We edited the title to identify the locality as follows: “Prevalence of psychological frailty in Japan: NCGG-SGS as a Japanese national cohort study.” We added an explanation of frailty by referring to Collard’s paper in the introduction section.

Comments

Introduction:

The introduction is clear, short and simple. It sets out and justifies the aim of the study. The literature review includes other research, but they are not enough. The authors should consider some recent publications about frailty and their risk factors. Only a few papers have been published in the last five years and recently, there have been a lot of new studies on frailty.

Response

We re-performed the literature review and introduced previous research on frailty in the introduction.

Comments

Experimental section:

If applicable, please provide the number of study protocol registered with the Bioethical Committee.

Response

We added the number of the study protocol registered with the Ethics Committee.

Comments

As to the inclusion and exclusion criteria, what about patients with chronic kidney disease? Were they in the study group? In end-stage renal disease, uremia symptoms are often accompanied by frailty. These symptoms are linked to biological aging and occur at an earlier age in chronic dialysis patients than in the general population, which strongly correlates with prognosis. For this reason, ESRD can be treated as a model of accelerated aging. There are many similarities between uremia stages and aging, so the obtained results from hemodialysis patients can be extrapolated to the population of older individuals.

Response

We examined the relationships between physical frailty and estimated glomerular filtration rate (eGFR) in our participants. A total of 3210 participants (78.3%) had 60 and over for the eGFR value (Table 1, Figure 1). The residual analysis demonstrated that the prevalence of frailty increased when the eGFR value was less than 60 (Figure 2). We don’t have information on dialysis patients, but only five subjects had a serum creatinine level of 3 or higher. The results of the sub-analyses suggest a relationship between CKD and frailty, as you have pointed out. However, if this finding is included in this study, readers may be confused, so I would like to summarize it as another research paper. Thank you for pointing this out.

Table 1. eGFR value of the participants

eGFR

Number

Percent

Effective percent

Cumulative percent

≥90

396

9.6

9.7

9.7

60-89

2814

68.2

68.7

78.3

45-59

763

18.5

18.6

96.9

30-44

105

2.5

2.6

99.5

15-29

16

0.4

0.4

99.9

<15

5

0.1

0.1

100

Total

4099

99.3

100

Missing

27

0.7

Figure 1. Histogram of eGFR of the participants

Figure 2. Prevalence of frailty by eGFR

Comments

A flow chart with recruitment of participants, drop-outs (deaths and others) with the final number of participants analyzed would facilitate the readability of the manuscript.

Response

We added the figure of the participants’ flow in the experimental section.

Figure 1. Participants’ flow and frailty status.

Comments

Were the known Katz Index [Katz S., Ford A.B., Moskowitz R.W., Jackson B.A., Jaffe M.W. Studies of Illness in the Aged. The Index of ADL: A Standardized Measure of Biological and Psychosocial Function. JAMA. 1963; 185: 914–919] and Lawton IADL Index [Lawton M.P., Brody E.M. Assessment of older people: Self-maintaining and Instrumental Activities of Daily Living. Gerontologist. 1969;9:179–186] used, their modifications or some other scales?  The system of classification is not transparent to the public. Authors need to be clearer. What are the mentioned LTCI (L. 122) and ADL or “Care Level 1 through 5” (L. 124) support scales? Would it be possible to provide them as supplementary data? Adding background information about LTCI and ADL should be considered as well. It is unclear which scale for the assessment of functional status was used.

Response

Unfortunately, we did not perform ADL measurements such as the Kats Index and Lowton Index. In this study, we used data from the long-term care insurance (LTCI) to which all older people belong, to prevent data loss of the primary outcome. Japan implemented the LTCI system on April 1, 2000, and every Japanese person aged 65 and older is eligible for benefits based strictly on their physical and mental frailty or disability.3 The LTCI is a mandatory national insurance plan for older people in Japan, and it was instituted to support the independence of older people, rather than simply providing personal care. The certification for LTCI has been reported in detail elsewhere.3 Briefly, all individuals aged 65 years or older, or those aged 40–64 years who suffer from age-related diseases are eligible for LTCI benefits in Japan. When a person applies to their municipality for LTCI benefits, an authorized care manager examines their physical and mental status using a standardized questionnaire. Then the certification board, which includes medical doctors and nurses, determines the level of long-term care necessary based on the estimated time required for care, as well as on comments from the applicant’s family physician. The LTCI certifications consist of the following eight levels: independent, support levels 1–2, and care levels 1–5. We defined a person certified as LTCI support level 1 as a person who can independently carry out basic activities of daily living but requires some assistance in instrumental activities of daily living, or higher as a disability. We added an explanation about LTCI to the text.         

Tsutsui T, et al. Care-needs certification in the long-term care insurance system of Japan. J Am Geriatr Soc. 53(3): 522-527, 2005.

Comments

The subsection Measurements of life style activity (L. 127–139) is also unclear. The citation 33 refers to the study of Shimada et al., which includes a scale that contains 16 questions. The scale used in this study consists of 15 questions. Why was the question “Do you talk with other people every day?” not included in the present study? Describe what was encompassed in the geriatric assessment. Was it a comprehensive geriatric assessment? The methods lack clarity and are in some parts confusing.

Response

The lifestyle activities were assessed using questionnaires that combine the items of previous research. We have added the answers to the questions to the text. The NCGG-SGS cohort study is a structural questionnaire survey that does not provide a comprehensive history of current disease, and it is the survey of only typical diseases as illustrated in Table 1.

Comments

Results:

The results are almost clearly formatted and presented. The units and other notations are correct. Graphs, axis heading, and data labels are readable. Table 1: The groups are defined in the headline of the table. Adjustment this long table for publication would certainly make it easier to read.

A good idea would be to use abbreviations and indicate on the bottom of the table: group R; group PhF, group D, group PF.

Response

We added an abbreviation in Table 1.

Comments

Discussion:

Like in the introduction, some recent publications should be integrated into the discussion section. Similar to the organizing info issue in the intro, list info about the previous findings in relations to the main findings of the present study. Organization of information regarding previous literature and the data interpretation especially clinical relevance of the findings could be improved. In literature prevalence of frailty is found higher than prevalence in the present study. Please try to explain it further. In such an important issue for geriatric medicine, the discussion is too short; please go a bit deeper with some more relevant reference. As the authors themselves note the level of independence in activities of daily living rather than life expectancy are of the utmost importance at this age. There is an increasingly wide literature about frailty worldwide. These issues should be more fully debated in the manuscript noting that the findings in this study represent a locality-based study which may/may not be replicated elsewhere. Substantial differences in the prevalence of frailty across countries are observed, with a steep increase with aging.

 Response

Thank you for your important comments. We re-performed the literature review and introduced previous research on frailty in the introduction and discussion sections. 

Added references

Collard, R.M.; Boter, H.; Schoevers, R.A.; Oude Voshaar, R.C. Prevalence of frailty in community-dwelling older persons: A systematic review. J Am Geriatr Soc 2012, 60, 1487-1492. Freiheit, E.A.; Hogan, D.B.; Eliasziw, M.; Meekes, M.F.; Ghali, W.A.; Partlo, L.A.; Maxwell, C.J. Development of a frailty index for patients with coronary artery disease. J Am Geriatr Soc 2010, 58, 1526-1531. Siriwardhana, D.D.; Hardoon, S.; Rait, G.; Weerasinghe, M.C.; Walters, K.R. Prevalence of frailty and prefrailty among community-dwelling older adults in low-income and middle-income countries: A systematic review and meta-analysis. BMJ Open 2018, 8, e018195. O'Caoimh, R.; Galluzzo, L.; Rodriguez-Laso, A.; Van der Heyden, J.; Ranhoff, A.H.; Lamprini-Koula, M.; Ciutan, M.; Lopez-Samaniego, L.; Carcaillon-Bentata, L.; Kennelly, S., et al. Prevalence of frailty at population level in European advantage joint action member states: A systematic review and meta-analysis. Ann Ist Super Sanita 2018, 54, 226-238. Kojima, G.; Iliffe, S.; Taniguchi, Y.; Shimada, H.; Rakugi, H.; Walters, K. Prevalence of frailty in japan: A systematic review and meta-analysis. J Epidemiol 2017, 27, 347-353. Edwards, J.D.; Lunsman, M.; Perkins, M.; Rebok, G.W.; Roth, D.L. Driving cessation and health trajectories in older adults. J Gerontol A Biol Sci Med Sci 2009, 64, 1290-1295. Choi, M.; Lohman, M.C.; Mezuk, B. Trajectories of cognitive decline by driving mobility: Evidence from the health and retirement study. Int J Geriatr Psychiatry 2013, 29, 447-453. Marottoli, R.A.; Mendes de Leon, C.F.; Glass, T.A.; Williams, C.S.; Cooney, L.M., Jr.; Berkman, L.F.; Tinetti, M.E. Driving cessation and increased depressive symptoms: Prospective evidence from the new haven epese. Established populations for epidemiologic studies of the elderly. J Am Geriatr Soc 1997, 45, 202-206. Freeman, E.E.; Gange, S.J.; Munoz, B.; West, S.K. Driving status and risk of entry into long-term care in older adults. Am J Public Health 2006, 96, 1254-1259. Edwards, J.D.; Perkins, M.; Ross, L.A.; Reynolds, S.L. Driving status and three-year mortality among community-dwelling older adults. J Gerontol A Biol Sci Med Sci 2009, 64, 300-305. Shimada, H.; Hotta, R.; Makizako, H.; Doi, T.; Tsutsumimoto, K.; Nakakubo, S.; Makino, K. Effects of driving skill training on safe driving in older adults with mild cognitive impairment. Gerontology 2018.

  Comments

Conclusions:

The conclusions were correctly presented. The observations of Shimada et al. confirm the multidimensional scope of frailty: a syndrome that includes both physical and psychological conditions. Conclusions support the concept of a psychological dimension of physical frailty in geriatric inpatients. The prognostic value of an assessment that accounts for both physical and mental components of frailty is superior to the separate values of these components. As mentioned above, local groups can vary widely. Thus, conclusions may not necessarily have universal range and need further studies.

Response

Based on your suggestions, we added the need for future research.

Comments

Minor issues:

78: The phrase ‘We excluded...’ should be capitalized 121: It should read ‘In this study’

Response

We revised the manuscript in accordance with your comments.
